# High Prevalence of Non-Responders Based on Quadriceps Force after Pulmonary Rehabilitation in COPD

**DOI:** 10.3390/jcm12134353

**Published:** 2023-06-28

**Authors:** Marion Desachy, François Alexandre, Alain Varray, Virginie Molinier, Elodie Four, Laurène Charbonnel, Nelly Héraud

**Affiliations:** 1EuroMov Digital Health in Motion, University Montpellier, IMT Mines Ales, Montpellier, France; alain.varray@umontpellier.fr; 2Direction de la Recherche et de l’Innovation en Santé (Research and Health Innovation Department), Clariane, France; francois.alexandre@clariane.fr (F.A.); virginie.molinier@clariane.fr (V.M.); nelly.heraud@clariane.fr (N.H.); 3Clinique du Souffle Les Clarines, Inicea, France; elodie.four@inicea.fr (E.F.); pradines.laurene@wanadoo.fr (L.C.)

**Keywords:** chronic disease, rehabilitation, muscle force

## Abstract

Pulmonary rehabilitation (PR) in patients with COPD improves quality of life, dyspnea, and exercise tolerance. However, 30 to 50% of patients are “non-responders” (NRs) according to considered variables. Surprisingly, peripheral muscle force is never taken into account to attest the efficacy of PR, despite its major importance. Thus, we aimed to estimate the prevalence of force in NRs, their characteristics, and predictors of non-response. In total, 62 COPD patients were included in this retrospective study (May 2019 to December 2020). They underwent inpatient PR, and their quadriceps isometric maximal force (Q_MVC_) was assessed. The PR program followed international guidelines. Patients with a Q_MVC_ increase <7.5 N·m were classified as an NR. COPD patients showed a mean improvement in Q_MVC_ after PR (10.08 ± 12.97 N·m; *p* < 0.001). However, 50% of patients were NRs. NRs had lower pre-PR values for body mass, height, body mass index, PaO_2_, and Q_MVC_. Non-response can be predicted by low Q_MVC_, high PaCO_2_, and gender (when male). This model has a sensitivity of 74% and specificity of 81%. The study highlights the considerable number of NRs and potential risk factors for non-response. To systematize the effects, it may be interesting to implement blood gas correction and/or optimize the programs to enhance peripheral and central effects.

## 1. Introduction

Pulmonary rehabilitation (PR) is ratified by all current international guidelines as one of the most effective treatments for chronic obstructive pulmonary disease (COPD) [1]. It is a comprehensive intervention consisting mainly of exercise training (considered to be the cornerstone of PR), education, nutritional intervention, and psychosocial support [2]. Numerous studies have highlighted the effectiveness of PR in relieving dyspnea and symptoms of anxiety and depression, as well as in improving the exercise tolerance and health status of patients with COPD [1,3].

Nevertheless, many studies have reported significantly heterogeneous individual responses to PR [4,5,6,7]. For a given outcome, some patients achieve improvement following a program while others do not. Patients who do not exhibit an improvement higher than the minimal clinically important difference (MCID) are referred to as “non-responders” (NRs). NRs are found for all types of variables, either biological, psychological, or social, and they represent a disconcertingly high number of patients. Indeed, depending on the outcomes and studies considered, the number of NRs to pulmonary rehabilitation ranges between 30% to 50% of patients [4,6]. The existence of NRs has been demonstrated, in particular, for exercise tolerance (6 min walk distance test [6 MWD]), quality of life (St. George’s Respiratory Questionnaire [SGRQ]), and anxiety or depressive symptoms (Hospital Anxiety and Depression Scale [HADS]) [6,7,8,9].

Among the various studied outcomes, the existence of NRs in terms of muscle force has not been documented to date. However, muscle weakness is an extremely deleterious comorbidity that affects between 32% and 57% of COPD patients and exists at all stages of the disease [10]. It contributes to exercise intolerance [11] and increased functional disability [12] and is a major predictor of patient survival [13]. Despite this, there are currently no data concerning NRs in terms of muscle force following PR. Indeed, even if the literature reports moderate efficacy, individual responses have never been investigated, thus providing only a partial view of the efficacy of PR on muscle force. 

Taken together, these elements highlight that muscle force is a very important dimension to consider in COPD and also explain why improving force has progressively become a central objective of PR in international guidelines [14]. The current literature supports the efficacy of PR, reported as an average value, in improving force [15]. Unfortunately, by only reporting average values and not individual patient responses, previous studies provide only a partial view of the effectiveness of PR on muscle force.

Thus, the main objective of the present study was to estimate the prevalence of NRs in terms of muscle force after PR. In addition, the two secondary objectives were to define the characteristics of responder and NR patients and to identify predictors of a lack of force gain in the latter group. 

## 2. Materials and Methods

### 2.1. Study Design and Participants

The study was conducted in accordance with the French legislation concerning retrospective studies (National Committee for Computing and Liberties (CNIL), reference methodology n°004, Deliberation n° 2018-155 of 3 May 2018) and the protocol had been deposited on the French Hub Data Health platform before the beginning of the analyses. We first identified eligible patients based on medical records. A total of 1171 patients were considered eligible for the study due to their stay at the clinic “Clinique du souffle Les Clarines” between May 2019 and December 2020. Of these 1171 patients, 1070 did not undergo maximal force testing, largely because this test is ordered by a medical doctor only if there is a suspicion of muscle problems identified at the first visit. Of the remaining 101 patients who underwent force testing, 38 were not COPD patients (they had a Tiffeneau ratio >70) and 1 patient refused the reuse of his personal data. Finally, 62 patients were included in the study (Figure 1). 

### 2.2. Pulmonary Rehabilitation Program

All patients completed a four-week inpatient PR program whose content was in accordance with international guidelines [16]. Patients attended sessions of the program five days a week, consisting of exercise training (endurance, resistance training, and balance and motor skills training) and therapeutic education. More specifically, the force training program consisted of 3 sessions per week for 4 weeks (12 sessions), each lasting at least 45 min. In the exceptional situation where a patient missed a session during the week, they could attend a catch-up session on Saturday morning. If the patient missed several sessions, or was unable to attend the Saturday morning session, their stay was extended. Each session included four exercises: two exercises targeting the muscles of the lower limbs and two exercises targeting the muscles of the upper limbs. The lower-limb exercises included leg extension, adduction, abduction, ankle extensor, ankle flexor, and knee flexor exercises. The upper-limb exercises included elbow flexor, elbow extensor, shoulder depressor, scapula stabilizer, external rotation of the shoulder, and abdominal belt (transverse and obliques) exercises. All sessions were conducted in a group and supervised by an adapted physical activity teacher or physiotherapist.

The force training sessions were mandatory, and the presence of patients was monitored. 

During the sessions, patients were encouraged, provided with positive feedback, and were able to track their progress, which motivated them.

In terms of progression, all patients worked at an RPE of 7 on a visual analog scale ranging from 0 to 10, corresponding to an intensity of between 70 and 80% of RM [17]. This intensity follows international ATS/ERS recommendations of 8–12 RM, which corresponds to 69.4–80.5% of 1 RM according to Brzycki’s equation [18,19]. They gradually increased the number of repetitions, starting with three sets of ten reps, then three sets of twelve reps, and finally three sets of fifteen reps. They had 1 min and 30 s of rest between sets and were required to maintain a tempo of three seconds in concentric and eccentric phases.

Although the exercises were generic, using either machines, resistance bands, or bodyweight alone, the sessions were individualized based on the participants’ RPE for the weights and the elastic bands. This individualization allowed participants to progress at their own pace, making the program accessible to people with different levels of fitness and ability.

### 2.3. Clinical Features

Patient characteristics such as sex (male or female), age (years), body mass (kg), muscle mass index (i.e., lean body mass divided by height^2^: FFMI in kg_LBM_/m^2^), and body mass index (BMI in kg/m^2^) were collected. 

Lung function was evaluated using plethysmography (V6200 Autobox, SensorMedics Corp., Yorba Linda, CA, USA) according to the ATS/ERS guidelines [16]. The forced expiratory volume in one second (FEV_1_) was expressed in liters and as a percentage of the predicted value. The Tiffeneau ratio was calculated by dividing the measured FEV_1_ by the measured forced vital capacity (FVC).

Measurement of blood gases was performed to determine the partial pressure of oxygen (PaO_2_ in mmHg) and the partial pressure of carbon dioxide (PaCO_2_ in mmHg). This measurement was performed on the radial artery in resting patients while they breathed room air and was conducted using a blood gas analyzer (ABL 825, Radiometer Medical, Bronshoj, Denmark).

Exercise tolerance was determined by the distance (in m) walked during a 6 min walk test (6-MWT) according to ERS/ATS technical standards before and after PR [20]. 

Patients completed the VQ-11 questionnaire [21] before and after the PR. This was used to assess quality of life.

### 2.4. Quadriceps Maximal Voluntary Force

Muscle force was measured during maximal voluntary isometric contraction of the quadriceps (Q_MVC_) of the right leg before and after the PR program. The participants were seated on a dedicated ergometer for knee extensor testing (Quadriergoforme, Aleo Industrie, Salome, France) equipped with a strain gauge torque sensor (Captels, Saint Mathieu de Treviers, France). The hip and the knee angle were set at 90°. The pelvis and the proximal extremity of the patella were securely attached to the chair in order to minimize the movement of adjacent muscles. To ensure localized force production in the quadriceps, the upper limbs were bent and secured to the thorax. After a short warm-up consisting of repeating several submaximal contractions with visual feedback of the torque on a screen, the patients had to perform three maximal voluntary contractions for 3 s. The best of the three trials was recorded as the maximal quadriceps torque. If the assessor observed a variation of more than 10% between the three trials, patients were asked to perform two additional maximal voluntary contractions. Assessment of muscle force before the PR program, expressed as a percentage of the predicted value, was used to identify patients with muscle weakness. A patient was considered to have muscle weakness if they exhibited a Q_MVC_ value lower than 80% of the predicted value [22].

The measurement of muscle force before and after the stay, expressed in N·m, enabled the identification of patients who did not respond to the PR program in terms of muscle force. Patients who exhibited an improvement of less than 7.5 N·m (considered as the minimally clinically significant difference for muscle force [MCID [23]]) after PR were classified as non-responders (NRs). 

### 2.5. Statistical Analysis

All results are presented as their means ± the standard deviation (SD) in the case of normally distributed data, the median [25–75% percentile] in the case of abnormally distributed data, or as percentages when appropriate (for prevalence results). 

The effect of PR on the entire group was assessed using Student’s paired *t*-tests (or a Wilcoxon signed-rank test if the normality of the differences was not verified by the Shapiro–Wilk test). 

Characteristics between responder and non-responder groups were compared using Student’s unpaired *t*-tests (or the Mann–Whitney test in case of non-normal distribution in at least one group). The chi-squared test was used to compare the sex ratio, the prevalence of muscle weakness, and the ratio of long-term oxygen therapy (LTOT) patients in each group.

A multiple logistic regression analysis was carried out to identify factors associated with the probability of non-response after the program (i.e., a pre-PR to post-PR Q_MVC_ increase <7.5 N·m). To select the variables and find the best model in terms of sensitivity, specificity, and total prediction quality, the “all possible subset selection method“ was used [24]. This method examined all possible combinations of variables (age, sex, height, body mass, BMI, FEV_1_, FEV_1_/CVF, PaO_2_, PaCO_2_, FFMI, 6 MWD, Q_MVC_) to determine the best subset for the prediction model. We constructed models with one variable, two variables, three variables, and so on to determine which combination performs best according to specific criteria. Our specific criteria were sensitivity, specificity, and the ability to accurately predict non-response. To verify the quality of the model, we checked the potential theoretical association of any variable with non-response after PR, the likelihood ratio test, Pearson goodness-of-fit tests, and the variation inflation factor (VIF) to verify the absence of colinearity (5 was the maximum value accepted), the evolution of Akaike Information Criterion (AIC), and the normality of the residual distribution. 

The significance of odds ratio was assessed using the Wald test. To evaluate the quality of the model, a confusion matrix was created. This matrix showed the predicted values from the model vs. the actual values from the test dataset. It provided the sensitivity of the model (the “true positive” rate, which is the percentage of individuals who the model correctly predicted as NRs) as well as its specificity (the “true negative” rate, i.e., the percentage of individuals that the model correctly predicted as “responders”).

A threshold of 0.05 was considered significant for all statistical tests. Statistical analyses were performed by using JASP software [JASP Team (2022), JASP (Version 0.16.3)].

## 3. Results

### 3.1. Patient Characteristics

Patient’s characteristics are given in Table 1. 

### 3.2. Prevalence and Characteristics of NRs

The pooled COPD patients exhibited a significant improvement in their mean muscle force (10.1 ± 12.9 N·m; *t* = −6.1; *p* < 0.001, statistical power: 0.99) with a baseline Q_MVC_ value of 93.4 ± 34.6 N·m and a post-Q_MVC_ value of 103.5 ± 38.9 N·m. However, 50% of all patients (31 out of 62) were classified as NRs in terms of muscle force after PR (increase < 7.5 N·m) (Figure 2).

The NRs had significantly lower height, body mass, Q_MVC_, FEV_1_, and resting PaO_2_ and were more likely to require LTOT than the responders (*p* < 0.05). Interestingly, the number of patients with muscle weakness was very much the same between the groups (24 vs. 22 in the NRs and the responders, respectively). The complete characteristics of the patients within each group are presented in Table 2.

### 3.3. Predictors of Non-Response

The most predictive accurate model to predict muscle force non-response included three variables: sex, PaCO_2_ and Q_MVC_ (Table 3). Among them, two were independent predictors of non-response (PaCO_2_ (OR = 1.07; CI = 1.0–1.15) and Q_MVC_ (OR = 0.97; CI = 0.94–0.99)), while sex was added because it improved the prediction accuracy of non-response (sensitivity increased from 71 to 74.2% and specificity from 74.2 to 80.6%) (Table 4).

The probability of non-response increased when resting PaCO_2_ values were high, when Q_MVC_ at the beginning of the stay was low, and when the participant was male.

All these variables together resulted in the following equation:(1)$$\mathrm{Probability}\;\mathrm{of}\;\mathrm{NR}=\frac{1}{1+e^{-(-0.324\;+\;0.069\;\times \;{PaCO}_{2}-0.031\;\times \;QMVC\;+\;0.773\;\times \;sex)}}$$
with sex = 1 for men and 0 for women, PaCO_2_ in mmHg, and Q_MVC_ in N·m.

When the value of the non-response probability is greater than 0.5, the model will classify the participant as an NR. For example, for a man with a resting PaCO_2_ of 36.7 mmHg and a Q_MVC_ of 69 N·m, the probability of non-response is 0.70, and he will therefore be classified as an NR by the equation. A confusion matrix showing the quality of the model is provided below (Table 3).

## 4. Discussion

The aims of the present study were to quantify the prevalence of NRs in terms of muscle force after a PR program, to identify the characteristics of these patients, and to determine the predictors of non-response. Our main results show that half of the patients did not reach the MCID regarding muscle force. Secondary results show that NR patients had significantly lower height, muscle mass, and PaO_2_ compared to those in the responder group. A logistic regression model including muscle force at entrance, resting arterial capnia, and sex predicted non-response with a sensitivity of 74% and a specificity of 81%.

From a clinical point of view, the fact that one out of two patients is an NR is a disconcerting result and cannot be attributed to a lack of adherence to and/or a non-observance of the training program guidelines. Indeed, the program was provided on an inpatient basis, and the systematic participation of patients was controlled. Furthermore, the exercise training was conducted according to the ERS/ATS international guidelines [25]. Aside from these considerations, it is important to note that the mean quadriceps force improvement (+10.09 N·m) was significant and consistent with the data in the literature [15,26,27]. Taken together, these elements show that our training program was as efficient as in previous published studies and yielded equivalent mean results. However, our study is the first to quantify the individual responses in terms of quadriceps force in relation to the MCID, and it provides clear evidence that the prevalence of NRs is fully obscured by the mean values, which are influenced by half of the patients improving their muscle force. As a consequence (even in a context where the use of a parametric statistic was appropriate, as was the case here), this result highlights that the mean value of muscle force is an inappropriate measurement for determining the clinical effectiveness of PR. Indeed, the amplitude of the improvement in the responders influenced the mean sufficiently to induce a significant statistical result.

To avoid obscuring such a phenomenon, it is therefore necessary to use other ways to express the effects of rehabilitation. Use of the median might be more appropriate because it is a more robust statistical value, i.e., it is influenced less by the responder values [28,29]. A good illustration of this statement can be observed in our cohort, as we found a mean increase of +10 N·m but a median increase of only +2 N·m (i.e., 50% of the patients had a gain greater than 2 N·m and 50% had a gain of less than 2 N·m). Another complementary approach could be to systematically indicate the number of people with changes greater than the MCID in order to obtain a more relevant view of the cohort, as well as of the individual and clinical data.

Due to the major deleterious consequences of muscle weakness in COPD patients [11,12,13], the understanding of non-response is crucial to be able to systematize the efficacy of strengthening programs.

First, we identified higher PaCO_2_ as a significant predictor of non-response and, in the NR group, significantly lower basal PaO_2_. Several studies are consistent with such a specific blood gas status and with the existence of adverse effects from usual training adaptations. Hypercapnia is notably known to trigger a skeletal muscle atrophy pathway via activation of AMPKα2, phosphorylation of FoxO3a, and induction of MuRF1 [30,31,32]. Moreover, lower PaO_2_ is implicated in different muscle atrophy pathways. Indeed, chronic hypoxic stress causes a disturbance to muscle cell homeostasis with consequences on muscle metabolism, phenotype, and growth. Hypoxia is known to inhibit muscle anabolism (particularly via inhibition of the mTOR pathway [33,34,35,36]) and to increase catabolism pathways (autophagy, ubiquitin–proteasome system, etc. [37,38]). As a consequence, chronic hypoxemia increases the imbalance between protein synthesis and degradation, which explains the predominance of catabolic pathways and subsequent muscle atrophy. Taken together, these elements are consistent with the absence of force improvement after training in NR patients, which could be explained by the activation of different atrophy pathways counteracting the expected training effects. Aside from the perturbation of different muscle atrophy pathways, abnormal blood gases could lead to disorders of the central nervous system. Indeed, the literature indicates that hypercapnia induces an acute depressing effect on cortical function that is characterized by decreased cerebral oxygen consumption and neuronal activity [39,40]. More long-term, chronic hypoxemia (more frequent in NRs) may lead to deleterious effects on the brain via a higher risk of nocturnal desaturation [41]. When this occurs during a specific phase of sleep (non-rapid eye movement phase), markers of brain injury have been shown to be significantly increased [42], which means that the repetition of nocturnal desaturations can lead to anatomical lesions. Thus, the conjunction of higher PaCO_2_ and lower PaO_2_ support hypotheses of acute depressing cortical activity and chronic deleterious effects on brain integrity, respectively, which explain inadequate motor cortex activity during voluntary force production. As a consequence, any action focused only on muscle structure should be logically ineffective in COPD patients in this context [42].

Secondly, the high prevalence of NRs casts doubts on the nature of strengthening programs used. 

Indeed, the increased knowledge regarding COPD muscle weakness pathophysiology highlights the existence of two main components implicated in lower voluntary force production: intrinsic muscle qualities [43,44,45,46] and impaired central motor command [47,48]. Unfortunately, the muscle-strengthening programs used in PR do not sufficiently take into account this updated knowledge. From the perspective of optimizing existing programs, resistance training is known to improve motor control [49]. However, this improvement is inhibited when strength training sessions are preceded in the same day by endurance training sessions [50]. Therefore, proper planning would prevent muscle strengthening-induced neural adaptations from being inhibited in COPD patients. Another possible adaptation of existing programs is to increase training intensities to target neural adaptations. A meta-analysis found that high-intensity resistance training protocols (>60% of 1 RM, the maximal load on a single repetition) improved 1 RM more significantly than low-intensity protocols (<60% of 1 RM) while inducing comparable muscle mass increase [51]. These results suggest that the higher force gains achieved following high-intensity protocols are primarily underpinned by neural adaptations. To optimize central adaptations, other training modalities, such as eccentric training, may also be considered. Indeed, it has been shown that muscle strengthening in the eccentric modality induces a disproportionate increase in force compared to the increase in muscle mass in COPD patients [52]. Because these adaptations occur without structural changes to the muscle, these results suggest the existence of neural adaptations. These findings are consistent with the high demand on the nervous system observed when performing eccentric contractions [53]. Muscle electrostimulation is another means of soliciting the nervous system. Indeed, it allows for an increase in the level of voluntary activation concomitant with the muscle mass gain resulting from repeated muscle contractions [54]. Activation of peripheral afferent nerve fibers by muscle electrostimulation repeatedly stimulates sensorimotor and motor brain networks and induces an increase in cortical excitability [55]. As a result, voluntary activation is increased. 

It therefore seems particularly relevant to specifically objectify and validate the effectiveness of these training methods in COPD patients compared to conventional training methods in order to reduce the number of NRs.

## 5. Conclusions

This study highlights the existence of a heterogeneous response following a PR program, with half of the participants not demonstrating improvements in terms of muscle force. This result highlights the ineffectiveness of current programs for a significant number of patients, which can be obscured by the use of mean improvement values. Thus, there is a clear need for future studies to more accurately express the effects of PR programs using relevant indicators (e.g., the median improvement or the proportion of patients reaching the MCID).

In addition, major challenges need to be addressed to adapt strengthening program management in order to systematize muscle force improvement. Although each of the predictors has a relatively low predictive potential (low OR), taken together they remain highly predictive (high sensibility and specificity of the model). Thus, the need to correct blood gases during the conception of strengthening training dedicated to maximizing muscle and neuronal effects appears to be of prime importance. 

## 6. Methodological Considerations

It is important to note that this study had a retrospective design, which means that it is subject to recruitment and selection biases. In addition, the limited time period on which the study was conducted (May 2019–December 2020) may have also limited the number of eligible participants for inclusion, which may affect the generalizability of the results. The retrospective nature of the study prevented us from selectively choosing the data we collected, and certain information, such as program adherence, was not captured and therefore unavailable to us. However, we know that patients completed a minimum of 12 sessions, and if not, they were excluded from the program. Furthermore, considering that our force gains (means and standard deviation) are in line with those reported in the literature, our program seems to be sufficiently representative [56,57,58,59,60]. A classic concern related to retrospective designs is the inability to calculate the a priori necessary sample size (NSS), which leads to the potential risk of an inconclusive result due to low statistical power. Nevertheless, the a posteriori statistical power obtained on our main outcome (the force before and after PR) was excellent (0.99). This power analysis provides some confidence in the study’s ability to detect significant effects.

A second consideration deals with the duration of the program. Four weeks could be considered as a quite short duration for a training program. Indeed, the time course of training adaptations could be different among individuals. In this case, the “non-responder” patients could in fact be “slow responder” patients. However, this bias is unlikely because the force gains found in our study are comparable to those found in the literature on longer programs [27,59,60,61,62].

Finally, the dissociation between training and evaluation modalities (dynamic vs. isometric) could question our results. One may wonder if an isometric protocol is the optimal way to assess the specific effects of dynamic resistance training. However, the isometric protocol is a gold standard used to assess the effect of PR on COPD muscle force [25] and isometric force gain is a major aim of PR, especially because of its relationship with patient survival [13]. In addition, isometric and anisometric forces are highly correlated [63,64]. Most importantly, a non-optimal muscle force assessment cannot explain why half of the patients exhibited a significant muscle force improvement.

## Figures and Tables

**Figure 1 jcm-12-04353-f001:**
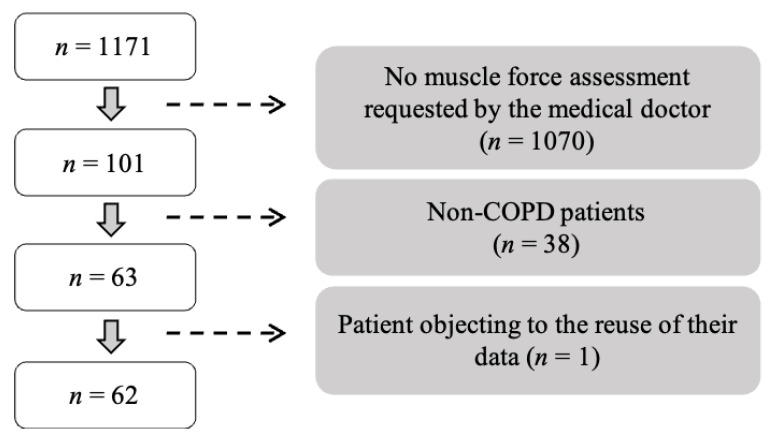
Number of patients screened, included, and excluded.

**Figure 2 jcm-12-04353-f002:**
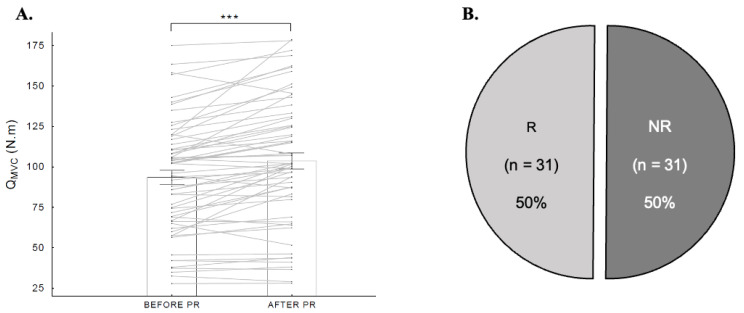
Effects of pulmonary rehabilitation on muscle force:(**A**) average increase in Q_MVC_ before and after pulmonary rehabilitation; (**B**) prevalence of non-responders (NR) and responders (R). Abbreviations: R: responders; NR: non-responders; Q_MVC:_ quadriceps maximal voluntary contraction. ***: *p* < 0.001.

**Table 1 jcm-12-04353-t001:** Baseline characteristics of the patients.

*n*	62
Male/female	41/21
Age (years)	64.1 ± 7.9
Body mass (kg)	72.5 ± 19.3
BMI (kg/m^2^)	25.8 ± 6.2
FEV_1_ (% predicted)	52.6 ± 21.2
FEV_1_ (L)	1.4 ± 0.6
6 MWD (m)	448.5 ± 91.5
Q_MVC_ (N·m)	93.4 ± 34.6
Q_MVC_ (% predicted)	73.4 ± 17.9

Abbreviations: BMI, body mass index; FEV_1_, forced expiratory volume in one second; Q_MVC_, quadriceps maximal voluntary contraction; 6 MWD, distance covered during the 6 min walking test.

**Table 2 jcm-12-04353-t002:** Characteristics of non-responder vs. responder COPD patients.

	Non- Responders (*n* = 31)	Responders (*n* = 31)	Mean between-Group Difference	95% CI
**Anthropometric parameters**				
Sex (M/F)	19/12	22/9	/	/
Age (years)	63.9 ± 8.0	64.3 ± 7.9	0.35	[−3.71; 4.42]
Body mass (kg)	65.9 ± 20.7	79.1 ± 15.5	13.16	[3.86; 22.47]
Height (cm)	165.2 ± 7.1	169.2 ± 7.6	4	[0.25; 7.75]
BMI (kg/m^2^)	23.9 ± 6.4	27.7 ± 5.5	3.81	[0.78; 6.84]
FFMI (kg/m^2^)	16.8 ± 2.5	17.7 ± 1.9	0.86	[−0.28; 2.02]
**Pulmonary parameters**				
FEV_1_ (L)	1.25 ± 0.63	1.56 ± 0.11	0.31	[−0.00; 0.62]
FEV_1_ (% predicted)	49.1 ± 22.8	56.1 ± 19.2	7.03	[−3.70; 17.76]
FEV_1_/FVC (%)	47.9 ± 11.3	52.9 ± 10.6	4.95	[−0.62; 10.53]
LTOT patients (%)	58	19	/	/
**Resting blood gases (room air)**				
PaO_2_ (mmHg)	60.6 ± 13.2	66.5 ± 8.9	5.90	[0.13; 11.64]
PaCO_2_ (mmHg)	40.2 ± 11.5	37.6 ± 4.4	−2.60	[−7.03; 1.82]
**Exercise capacity and QoL**				
6 MWD (m)	430 ± 17	466 ± 15	36.16	[−9.75; 82.07]
VQ11 score	32 ± 9	29 ± 11	−2.7	[−9.20; 3.79]
**Quadriceps force**				
Q_MVC_ (N·m)	83.1 ± 38.4	103.8 ± 27.2	20.81	[3.89; 37.73]
Q_MVC_ (% predicted)	71.0 ± 19.4	75.8 ± 16.2	4.77	[−4.32; 13.87]
ΔQ_MVC_ (N·m)	0.5 ± 5.7	19.7 ± 10.8	19.22	[14.81; 23.64]
Prevalence of muscle weakness (%)	77	71	/	/

Abbreviations: BMI, body mass index; FFMI, fat-free mass index; FEV_1_, forced expiratory volume in one second; PaO_2:_ partial pressure of oxygen in arterial blood; PaCO_2_: partial arterial carbon dioxide pressure; 6 MWD, distance covered in the 6 min walking test; Q_MVC_, quadriceps maximal voluntary contraction, LTOT: long-term oxygen therapy; QoL: quality of life; FVC: forced vital capacity; VQ11: questionnaire used to assess quality of life.

**Table 3 jcm-12-04353-t003:** Regression logistic model including Q_MVC_, PaCO_2_, and sex.

	Odds Ratio	95% IC	VIF
PaCO_2_	1.07	[1.0–1.15]	1.16
Sex	0.97	[0.50–9.4]	1.60
Q_MVC_	0.97	[0.94–0.99]	1.79

Abbreviations: PaCO_2_: partial arterial carbon dioxide pressure; Q_MVC_, quadriceps maximal voluntary contraction, VIF: Variance Inflation Factor.

**Table 4 jcm-12-04353-t004:** Confusion matrix for the predictive model of non-response including PaCO_2_, sex, and Q_MVC_.

		Predicted		
		Responders	Non-Responders	% Correct
**Observed**	Responders	25	6	80.6
	Non-responders	8	23	74.2
	Overall % correct			77.4

## Data Availability

The datasets used and/or analyzed during the current study are available from the corresponding author on reasonable request.

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
