# Peer review of "High Prevalence of Non-Responders Based on Quadriceps Force after Pulmonary Rehabilitation in COPD"

_jcm, 2023, doi:10.3390/jcm12134353_

Round 1

Reviewer 1 Report

The aim of this study was to estimate the prevalence of non-responders in terms of muscle strength after Pulmonary rehabilitation. In addition, two secondary objectives were to define the characteristics of responder and non-responder patients and to identify the predictors of the lack of strength gain for the latter. The authors concluded that half of the patients did not improve muscle strength. The study is retrospective in nature and there are several major issues that need to be addressed. My major comments and minor suggestions are listed below.

Major comments

The authors stated that “All patients completed a four-week inpatient PR program whose content was in accordance with international guidelines”. However, there is no citation for the international guidelines. Further, it is stated that all patients worked with the RPE = 7. Was it the RPE that ranges between 6 and 20 or the scale that ranges between 0 and 10? If the first was used, the intensity was too low. If the latter was used, the intensity is higher than what is recommended in clinical guidelines. Last, the patients performed 2 sessions of resistance training per week (for four weeks). Is this enough to produce gains in peripheral muscle force (the median increase was only 2 N.m)? Most pulmonary rehabilitations programs offer between 8 and 12 weeks of exercise (which includes resistance and aerobic training), 2 to 3 times per week (i.e. at least double of the number of sessions offered in the inpatient program in the current manuscript).

Regarding statistical analysis and reporting of the results, why did the authors choose to use standard error of the mean? My suggestion is to present data that are normally distributed as mean and standard deviation and data that do not have a normal distribution as median [25% - 75% percentile]. Further, when comparing groups (responder versus non-responder), the authors should present the mean difference and 95% confidence interval both in the table and in the text (rather than t value and p value). And any between-group difference should be interpreted based on the 95% confidence interval (rather than p value) and the word “significant” when talk about differences should be avoided. I suggest the authors to read the following work: https://www.tandfonline.com/doi/full/10.1080/00031305.2018.1527253. Last, when presenting odds ratio and 95% CI, p value is not needed.

It is arguable that PaCO2 is an independent predictor of response to pulmonary rehabilitation as there was no between-group difference between responders and non-responders (table 2) in PaCO2. Why was PaCO2 included in the regression models if there was no between-group difference in this variable? Only variables that were different between the groups should be included in the regression models.

Another major issue is that intervention fidelity (compliance, adherence and completion) was not reported. Although the program was an inpatient program, and usually compliance and completion is better than outpatient programs, the authors must report at least number of sessions of resistance training completed, any adverse event, reasons for patients not completing a session, any issue with progression of resistance training prescription and so on. Then, the authors should compare adherence/compliance and completion between responders and non-responders.

Minor suggestions

Overall:

-        Muscle strength versus muscle torque versus muscle force. The authors should be consistent with the use of the term. What was measured was actually muscle force (this would be the best term in the English language).

Abstract:

- The abbreviation NR is not explained

- Mean improvement is presented without standard deviation and Odds Ratios are reported without 95% CI.

- “Inclusion of sex improved non-response detection”. The information missing is: does female sex or male sex improve non-response detection?

Introduction:

-        Clear and concise

Methods:

-        Line 81: the sentence has duplicate information and should be omitted: “Ultimately, one patient objected to the reuse of their data, and a total of 62 individuals with COPD were included in the study.”

-        What is a “physical activity professional”?

-        Replace “progressiveness” with progression.

Conclusion:

-        Sentence “With half of the patients not improved”. Change to: With half of the participants not demonstrating improvements in…

The English language throughout the manuscript is fair. A revision by a native english speaker would improve the quality and is definitely suggested.

Reviewer 2 Report

My comments:

                In general, this is an interesting research. The authors aimed to estimate the prevalence of non-responders, their characteristics and predictors of non-response. They found that half of the patients not improved in terms of muscle strength. Moreover, they also found that the most predictive for non-response of muscle strength were base line PaCO2 and QMVC. However, there are some points should be improved.

Major concern

                The results of multivariable logistic regression must be shown in table format. According to your results in the table 2, many variables were significantly different between non-responders and responders groups especially in body weight and BMI. Body weight and BMI could affect the muscle strength. Moreover, others baseline characteristics including FEV1, LTOT, and PaO2 were also significant difference between groups. Thus, all of factors that different between groups must be accounted in the multivariable regression analysis.  I thin

Minors

2. Materials and Methods

            In line 114, the reference of ATS/ERS guidelines should be provided.

                In line 122, the 6-MWD should be change to 6-minute walk test (6-MWT).

3. Discussion

            The discussion section should be changed according to new results from multivariable regression analysis.

4. Conclusion

            The conclusion section should be changed according to new results from multivariable regression analysis.

5. References

            All references should be changed according to MDPI style.

Round 2

Reviewer 1 Report

Thank you for the comprehensive responses and changes.

Table 2: Rather than including 95% CIs for the within-group values, the authors should omit those and include the between-group difference and 95% CI. Both t values and p values should be avoided.

The authors have now stated that "Several combinations of variables were included in the model, included those that showed significant differences between the two groups (age, sex, height, body mass, BMI, FEV1, FEV1/CVF, PaO2, PaCO2, FFMI, TDM6 distance, QMVC pre and post PR)." This information is incorrect as many of these variables were not different between the groups (and TDM6 distance is not what was included in the table - 6MWD?).

The key to a successful logistic regression model is to choose the correct variables to enter into the model. While it is tempting to include as many input variables as possible, this can dilute true associations and lead to wide and imprecise confidence intervals, or, conversely, identify spurious associations. The conventional technique is to first run the univariate analyses (i.e., relation of the outcome with each predictor, one at a time) and then use only those variables which meet a preset cutoff for significance to run a multivariable model. Is this case, as between-group differences have been investigated, only BMI (as height and weight are used to calculate BMI), LTOT, PaCO2 and Qmvc should be included in the multivariate logistic regression.

Reviewer 2 Report

All comments were appropriately respond by the authors. This manuscript can be accepted.

Author Response

We warmly thank the Reviewer 2 for his positive comments and we are happy that our revisions were relevant and improved the manuscript.